

# Fitness consequences of fish circadian behavioural variation in exploited marine environments

Martina Martorell-Barceló[1],*, Andrea Campos-Candela[1,2],* and Josep Alós[1],*

[1] Instituto Mediterráneo de Estudios Avanzados, IMEDEA (CSIC–UIB), Esporles, Spain
[2] Universidad de Alicante, Alicante, Spain
* These authors contributed equally to this work.

## ABSTRACT

The selective properties of fishing that influence behavioural traits have recently gained interest. Recent acoustic tracking experiments have revealed between-individual differences in the circadian behavioural traits of marine free-living fish; these differences are consistent across time and ecological contexts and generate different chronotypes. Here, we hypothesised that the directional selection resulting from fishing influences the wild circadian behavioural variation and affects differently to individuals in the same population differing in certain traits such as awakening time or rest onset time. We developed a spatially explicit social-ecological individual-based model (IBM) to test this hypothesis. The parametrisation of our IBM was fully based on empirical data; which represent a fishery formed by patchily distributed diurnal resident fish that are exploited by a fleet of mobile boats (mostly bottom fisheries). We ran our IBM with and without the observed circadian behavioural variation and estimated selection gradients as a quantitative measure of trait change. Our simulations revealed significant and strong selection gradients against early-riser chronotypes when compared with other behavioural and life-history traits. Significant selection gradients were consistent across a wide range of fishing effort scenarios. Our theoretical findings enhance our understanding of the selective properties of fishing by bridging the gaps among three traditionally separated fields: fisheries science, behavioural ecology and chronobiology. We derive some general predictions from our theoretical findings and outline a list of empirical research needs that are required to further understand the causes and consequences of circadian behavioural variation in marine fish.

# INTRODUCTION

Humans have exploited fish populations through trait-selective harvesting since the origin of our species (*Allendorf & Hard, 2009*). In fact, fishing is widely recognised today as a major driver of contemporaneous evolution and trait change in wild fish populations (*Sullivan, Bird & Perry, 2017*). There is substantial evidence that size-selective harvesting

Corresponding author
Josep Alós, alos@imedea.uib-csic.es

(e.g. gear selectivity) usually selects for fast life-histories and favours early maturation and high reproductive investment (*Alós et al., 2014*; *Heino, Pauli & Dieckmann, 2015*; *Laugen et al., 2014*; *Matsumura, Arlinghaus & Dieckmann, 2011*). The behavioural dimension of fisheries selection has recently gained interest among fisheries scientists and managers due to the growing evidence of consistent between-individual differences in the behaviour of exploited fish and the study of selection in real fisheries (*Arlinghaus et al., 2017*; *Diaz Pauli & Sih, 2017*, *Uusi-Heikkilä et al., 2008*). Currently, there is a large quantity of literature demonstrating the existence of consistent (in temporal and ecological contexts) between-individual differences of fish behavioural traits, such as boldness or aggressiveness, that define behavioural types within fish populations (*Conrad et al., 2011*; *Mittelbach, Ballew & Kjelvik, 2014*). In addition, with the recent development of aquatic telemetry, fisheries scientists have a powerful tool available to study behavioural types of free-living fishes (*Hussey et al., 2015*; *Lennox et al., 2017a*) and how fisheries may promote the selection of behavioural types in real-world fisheries (*Alós et al., 2016b*; *Monk & Arlinghaus, 2018*; *Olsen et al., 2012*). Together, these two developments have generated substantial empirical evidence demonstrating that bold and high-exploratory individuals (*Alós, Palmer & Arlinghaus, 2012*; *Biro & Sampson, 2015*; *Härkönen et al., 2014*; *Klefoth, Kobler & Arlinghaus, 2011*; *Olsen et al., 2012*) are more prone to harvest; thus, this evidence supports the idea that timidity syndrome can give rise to exploited fish populations that are composed of shy, less active and less exploratory individuals (*Arlinghaus et al., 2016*, *2017*).

Surprisingly, behavioural traits that determine timing have been poorly considered in the context of the selective properties of fishing. Recently, *Tillotson & Quinn (2017)* proposed the timing of migration or breeding as candidate traits that are targeted by fisheries selection. Both the timing of migration and the timing of the breeding season have strong impacts on population dynamics (*Lowerre-Barbieri et al., 2017*), and selection imposed by these traits would strongly impact the long-term trajectory of the fish stocks. Similarly, an ubiquitous behaviour related to timing in fish that has been overlooked by the scientific fisheries community is the manifestation of underlying circadian rhythms. Life on earth is governed by a 24-h rotation cycle that has led the evolution of endogenous circadian clocks across taxa, including fish species (*Kreitzman & Foster, 2005*). Similar to behavioural types, humans and some terrestrial animals show temporally consistent between-individual variation in different circadian-related behaviours, such as awakening time or sleep onset, that are the result of the interactions between those endogenous individual circadian clocks and the environment; furthermore, these interactions define chronotypes (*Roenneberg et al., 2007*; *Bloch et al., 2013*; *Rattenborg et al., 2017*). Although chronotypes should be ubiquitous across animal taxa, only a few studies have demonstrated the existence of chronotypes by exploring the amount of behavioural variation explained by between-individual differences (*Randler, 2014*), and these studies have mainly focused on bird species (*Dominoni et al., 2013*; *Steinmeyer et al., 2010*; *Stuber et al., 2014*, *2015*).

Regardless of whether fish sleep or not (*Reebs, 1992*), most fish species show a circadian-related behaviour in which they change from an active state to a resting state,

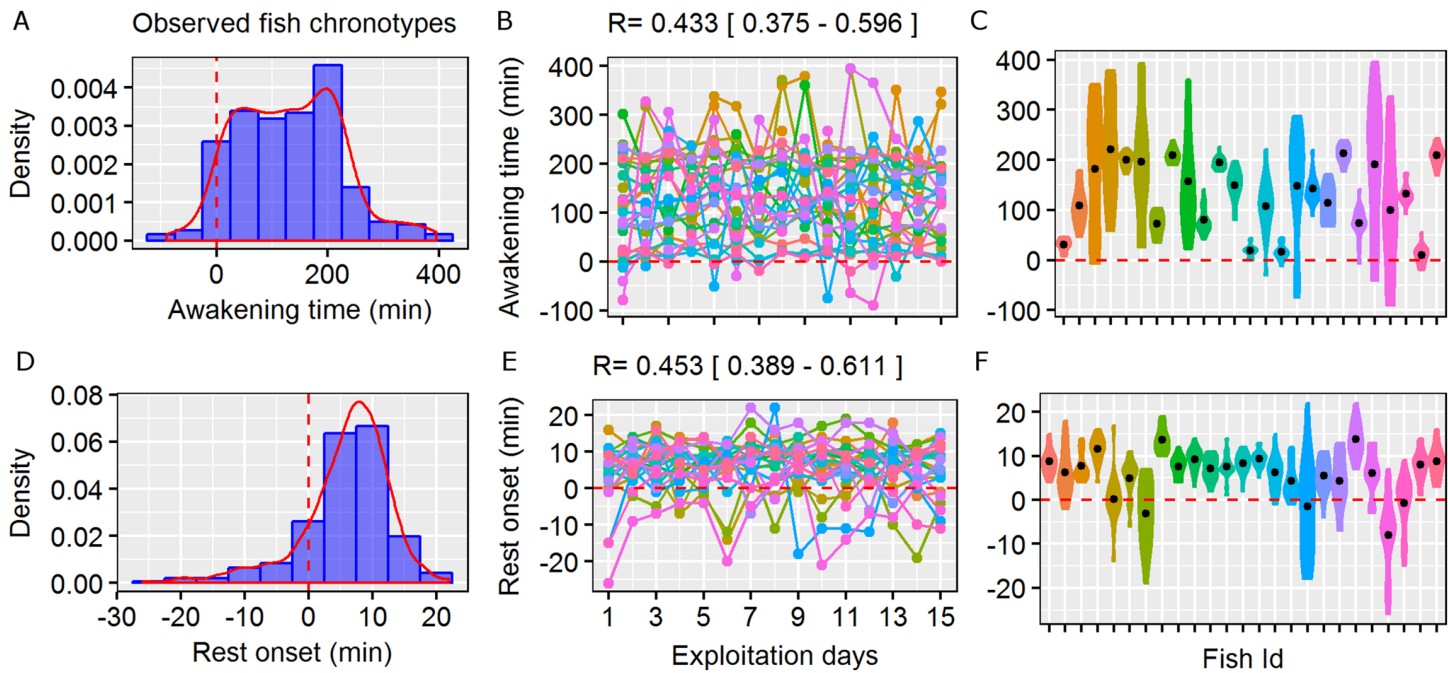

**Figure 1 Circadian behavioural variation in free-living marine fish.** Repeatability (*R*) of the wild behavioural variation in awakening time (moment of initiation of the active phase as minutes relative to the sunrise) and rest onset (moment of initiation of the resting phase as minutes relative to the sunset) observed in the pearly razorfish, *Xyrichtys novacula*. Sunset and sunrise are denoted by dashed red line. (A and D) Density and histogram plots showing the distribution in awakening time and rest onset from 25 randomly selected individuals from the simulated population. (B and E) Daily awakening time and rest onset (each colour represent a fish ID) across 15 days of simulated exploitation. The *R* scores and their confidential interval are plotted for each trait. (C and F) Individual violin plots showing the within- and among-individual variability (the individual mean is plotted as a black dot) in awakening and rest onset describing different types of chronotypes (e.g. early risers). Simulated data shown is based on the empirical work by *Alós, Martorell-Barceló & Campos-Candela (2017)*.

leading to a 'sleep-like' behaviour that is consistent with the sleep architecture observed in mammals (*Schmidt, 2014*; *Siegel, 2008*). This diel active/resting cycle is widely observed in free-living fish across species (*Krumme, 2009*; *Alós, Cabanellas-Reboredo & Lowerre-Barbieri, 2012*, *Alós et al., 2016b*; *Koeck et al., 2013*). Recently, *Alós, Martorell-Barceló & Campos-Candela (2017)* found the first evidence supporting the existence of chronotypes in fish focused on the pearly razorfish, *Xyrichtys novacula*. Similar to humans and birds, fish chronotypes arise from between-individual differences in circadian behavioural traits that are consistent over time and ecological contexts (Fig. 1). Far from being anecdotal, chronotypes have been frequently linked to many fitness processes in terrestrial animals, such as predation mortality or finding a reproductive mate (*Roenneberg, Wirz-Justice & Merrow, 2003*, and see review by *Adan et al., 2012*), and any directional selection pressure (i.e. either natural or human-induced) acting on chronotypes could lead to trait changes in terms of circadian behavioural rhythms (*Helm et al., 2017*). In fact, one recent study demonstrated how a potential environmental-induced change in a behavioural trait can influence circadian behavioural variation and impact fitness (*Dominoni et al., 2013*), i.e. city birds that started their activity earlier than their forest conspecifics highlighted that urban environments

(i.e. those with artificial lighting) can significantly modify biologically important rhythms in wild organisms and explained the potential reproductive advantages conferred to the early-rising birds in such an artificial environments. Similarly, we assumed here that early-riser fish chronotypes would be more vulnerable to fishing simply because the number of encounters between the fish and fishers was expected to be higher.

Based on this assumption, the objective of this work was to explore the plausibility of selection acting on fish chronotypes using a spatially explicit individual-based model (IBM). Our IBM assumed relatively simple movement rules that dictated the encounters between fish and fishers, and it was based on the real properties of a general bottom coastal fishery; additionally, the IBM explicitly incorporated social-ecological factors to add realism to our model (and simulations). The selection gradient ($S$), as a central measure of selection in traditional quantitative genetics with heritability (*Price, 1970*), has been widely used to describe trait changes in commercial and recreational fisheries (*Alós et al., 2016b*; *Monk & Arlinghaus, 2018*). We aimed here to estimate mean-standardised selection gradients on circadian behavioural traits to determine whether they were different from zero, and we compared them with previously reported gradients of other traits. Although the economic consequences of fisheries selection can be addressed by proper fisheries management (*Eikeset et al., 2013*), it can generate undesirable consequences in terms of ecosystem functioning (*Audzijonyte et al., 2013*; *Jørgensen et al., 2007*); specifically, this selection can notably reduce the recovery of overexploited stocks (*Uusi-Heikkilä et al., 2015*; *Walsh et al., 2006*) and decrease the recreational utility of fisheries (*Sutter et al., 2012*). Therefore, our final objectives were to make broader predictions about our findings and to stimulate research on the topic by providing a list of empirical research needs to fully disentangle the causes and consequences of fish chronotypes in exploited environments.

## MATERIALS AND METHODS

To explore whether fishing selection influences circadian behavioural traits, we developed a computational IBM where a fish population spatially behaves in a 2-D landscape and is exploited by a fleet of fishing boats during a fishing session (see Fig. 2 and video in SM1). Our IBM is spatially explicit because fish and fishers move (i.e. change position every minute) across the landscape according to different types of movement models. Encounters between the fish and fishers determined the mortality of the fish. Although encounters between fish and fishers do not always predict harvest (*Monk & Arlinghaus, 2018*), these encounters are among the most important components related to the vulnerability of most fishes (*Lennox et al., 2017b*), especially in bottom coastal fisheries (*Alós, Palmer & Arlinghaus, 2012*; *Alós et al., 2016b*). Our model was built under a prototypical bottom fishery where (i) target fish performed a sedentary behaviour that lead to the establishment of a home range (HR) area, (ii) the centres of activity were patchily distributed and formed a patchy landscape (which could be the consequence of a fragmented habitat), and (iii) fish were exploited by a fleet of mobile fishing boats. Our model was parametrised using empirical data from a popular recreational baited hook-and-line fishery located in Mallorca Island (Spain) that targeted pearly razorfish

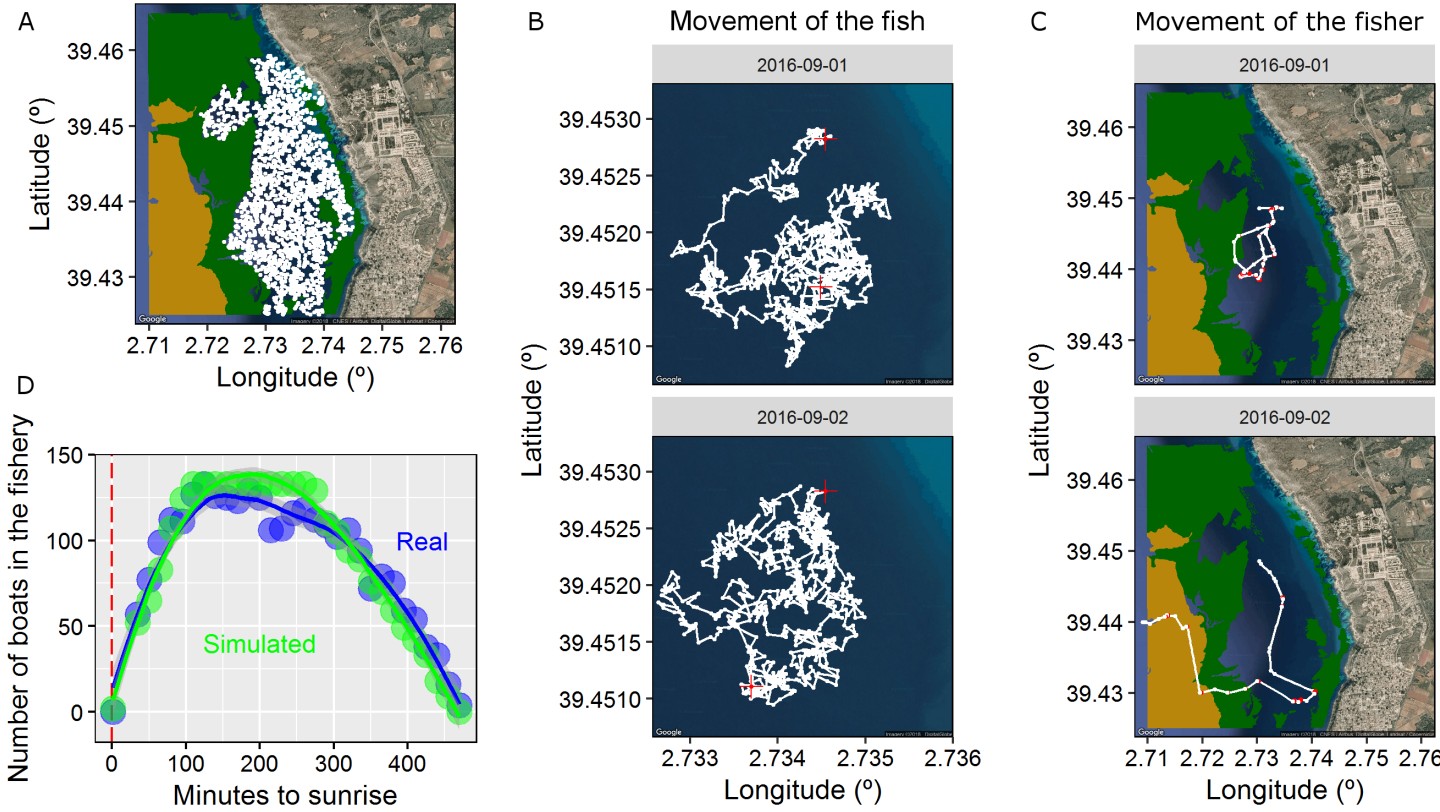

**Figure 2 Properties of the spatially-explicit individual based model (IBM) developed here.** (A) The 2-D landscape simulated here composed by different types of habitats (land, satellite seawater as the preferred fish habitat, seagrass in green and gravels in light brown). The centres of activity of each simulated fish (2,000 individuals) are shown in white. (B) Trajectory (positions every minute) of one fish in two different days. Red crosses represent the first and the last positions of the active diurnal phase. (C) Trajectory of one fisher in two different days. Red dots represent the positions were the fisher was fishing while white dots represents the positions were the fisher was searching according to the two-state movement pattern. (D) Number of boats in the virtual scenario every day aggregated in 15 min slots since the sunrise (the real data obtained using visual census is plotted in blue and the simulated data is plotted in green). The dashed red line represents the sunrise. The IBM was developed according to the real characteristics of the fishery developed in the waters of Mallorca Island (NW Mediterranean targeting the pearly razorfish, *Xyrichtys novacula*, *Alós et al., 2016b*).

(see full details in *Alós et al., 2016b*); however, the model is generalizable to any other system that displays these three main properties. Our computational IBM simulation was implemented and run in R (*R Core Team, 2017*). The R code is provided in the Supplementary Material (SM2).

### The ecological landscape: fish moving with individual heterogeneity in circadian and spatial behaviour

We created a 2-D landscape of 12.1 km$^2$ with open boundaries, of which 6.4 km$^2$ formed the preferred habitat of the pearly razorfish (hereinafter, the targeted species) to create a realistic ecological landscape (see map in Fig. 2). We randomly distributed 2,000 centres of activity (centre of the HR, see below) in the preferred habitat to create a patchy distribution of fish across the ecological landscape, and each centre of activity was designated to one identified fish (initial population = 2,000 individuals, density = 312

individuals per km², see Fig. 2). Then, fish survival was monitored every minute during the entire prototypical fishing season; here, survival was monitored for 15 full fishing days after the opening of the fishery on September 1st at 00:00, according to *Alós et al. (2016b)*. Thus, the IBM was discretised on time (every 1 min), had 21,600 time-steps ($n$), and the position (latitude and longitude) of each fish was mechanistically generated based on its movement and circadian behavioural variation, as described below.

Fish movement is usually mechanistically explained by different types of random walks (*Smouse et al., 2010*). Different from the purely random walks that generate standard diffusion across space, many fish species use a confined area and form stable HR areas (*Alós et al., 2016a*). The idea behind HR movement is that an individual moves within a harmonic potential field following random stimuli (i.e. a random walk); however, the individual has a general tendency to remain around a central place of residence (*Börger et al., 2006*). In such cases, there is a need for an additional behavioural rule that keeps the individual within its designated core site (*Benhamou, 2014*; *Smouse et al., 2010*), and this can be described by the Ornstein–Uhlenbeck process that defines a biased random walk (BRW) (*Alós et al., 2016a*).

For the purpose of this study, we focused on two descriptors of this BRW movement model described in *Alós et al. (2016a)*: (i) the size of the circular HR *radius* (in metres) that can be interpreted as a surrogate for the total foraging area and activity space, and (ii) the harmonic force ($k$, in min⁻¹) that can be interpreted as the strength of the drift or attraction force towards the centre of the HR, which ultimately determines the slope of the curve describing the cumulative space used in a period of time (we refer this as *exploration*). We randomly assigned values for both parameters to our virtual population of fish based on the real data estimated in *Alós et al. (2016a)*; range for *radius*: 67–470 m and *exploration*: 0.0005–0.025 min⁻¹, using the function `sample` of the base package of R. See Fig. 2 for a visualisation of the realised daily trajectories of a given fish.

Each of the 2,000 fish was assigned an individual mean and s.d. value for its awakening time and a daily value for its rest onset time based on the real data published in *Alós, Martorell-Barceló & Campos-Candela (2017)*; this generated the daily transition between the resting and active states at the individual level (see simulation scenarios below). Once a set of movement parameters and circadian behaviours was assigned to each identified fish, we generated a daily sequence of states (active vs. resting) based on the individual mean and s.d. values for each fish for the entire simulated fishing. Accordingly, we re-sampled the mean and s.d. of both circadian traits (i.e. awakening and rest onset times) daily for each individual, and we generated one value from this distribution for each day and individual (see Fig. 1). We then constructed the sequence of active and resting states based on the local sunset and sunrise times and the daily individual values that were generated. Finally, a position for all time-steps in an active state was generated for the entire fishing season based on our HR mechanistic model and the individual movement parameters of each fish (Fig. 2 and see SM1). During the resting state, the individual remained in the same position, and the fish was invulnerable to fishing as long as it remained in shelter (e.g. the pearly razorfish remains buried in the sand at night, according to *Alós, Cabanellas-Reboredo & Lowerre-Barbieri, 2012*).

The complete sequence of time-steps and positions for each fish was used to create a realistic dynamic ecological landscape (see movie SM1).

## The social landscape: a fleet of mobile boats targeting the ecological landscape

A fleet of mobile fishing boats exploited the ecological landscape. The entire fleet exploited the fishery every day during the entire fishing season (i.e. 15 days). On a daily basis, the IBM carefully considered different arrival and departure times for boats in the fishery (and the local sunrise data were used to synchronize the times with the ecological landscape), as this aspect is highly relevant for the objectives of our study. Specifically, we put effort into reproducing the real daily dynamics of fishing pressure by assigning a time of arrival and a time of departure for each boat (see Fig. 2), and these times were derived from a visual census of the actual fleet (*Alós et al., 2016b*); specifically, fishers exploited the fishery for a duration that ranged from 160 to 460 min after sunrise, with an effective fishing effort of 4.6 ± 1.2 h. For simplicity, no within-individual variability in the time of arrival and departure was considered (i.e. each fisher arrived at the fishery at the same time every day); however, some individuals arrived earlier than others, which is similar to the idea of fish chronotypes.

As fishers arrived at the fishery (depending on their individual arrival time), their spatial behaviour was based on a movement model that included two states. Individual boat fisher trajectories are usually composed of different states, and typically there are three main states: cruising, searching and fishing (*Vermard et al., 2010*; *Walker & Bez, 2010*). In our scenario, once fishers arrived at the fishery, they performed a classical search pattern that included two states, i.e. fishing and searching (see Fig. 2). Here, we considered relatively simple hidden Markov model (HMM) movement with two types of random walks describing each state (*Auger-Méthé et al., 2015*). HMMs are widely used for modelling any type of animal or fisher movement data (*Patterson et al., 2017*), and the R package *moveHMM* was recently developed to perform simulations of movement trajectories (*Michelot, Langrock & Patterson, 2016*).

Accordingly, for each fisher, a bi-variate time-series composed of step-lengths (in metre) and turning angles (in radius) was generated to describe the trajectory of each fisher every day. These temporal series were drawn by a state-dependent process at moment $n$ (unobserved in a real situation; the hidden Markov chain) using two distributions of the step-lengths and turning angles (one per state; fishing vs. searching). The transition among the two states was generated by a $2 \times 2$ transition probability matrix, $\Gamma = (\gamma_{ij})$, where $\gamma_{ij}$ was the probability of the fisher switching from the current state (at time-step $n$) to the future state (at time-step $n+1$). Here, we considered $\Gamma = \begin{pmatrix} 0.95 & 0.05 \\ 0.5 & 0.5 \end{pmatrix}$, meaning that each fisher spent most of his/her time fishing to obtain realistic fisher trajectories (see a realised trajectory of the fisher in Fig. 2).

Each state of the sequence was associated with a distinct random walk movement model that included a BRW for fishing and a correlated biased random walk (CBRW) for searching to adequately reproduce the spatial dynamics of the fleet (Fig. 2).

When the fisher was in the fishing state, the boat drifted with the current. Though this process is not a random walk, for simplicity, we used the mathematical description of a conventional BRW by biasing the angle of the trajectory according to the surface current in the area and adding some noise (see Fig. 2 and SM1). Accordingly, the step-lengths of this state were described by a gamma distribution (because velocity cannot obtain negative values), with the mean = 1 m and the s.d. = 0.5 m; additionally, the angle was described by a von Mises distribution, with the mean equal to the angle of the surface current and the concentration = 1.2 rad (noise) to reproduce similar real-life patterns observed in the fishery. To add realism, we used the real observed angle of the surface current for each time-step $n$ since September 1st, 2016 at 00:00; these data were obtained from an oceanographic buoy located in the study area by the SOCIB (www.socib.es) (*Tintoré et al., 2013*).

The searching state of the fisher was modelled using the CBRW model described by *Langrock et al. (2014)*, which was developed to model the group dynamics of animal movement. Accordingly, the searching state was mathematically described by a mixture of a BRW, where the bias was imposed by the social information that generated a tendency to move to the centroid of the positions of the other boats while searching (i.e. watching other boats, social information); and a conventional correlated random walk (CRW), where searching was described by a turning angle drawn from a von Mises distribution with a mean = 0 and a concentration = 5 rad. In both cases, the step-lengths were described by a gamma distribution of step-lengths with mean = 150 m and s.d. = 130 m (i.e. searching velocity). The BCRW developed by *Langrock et al. (2014)* is unique due to the existence of a parameter ($\eta$) that specifies the weight of the BRW with respect to the CRW portion of the BCRW. Here, we considered $\eta = 0.7$, which generated a behaviour of the fleet characterised by the tendency to remain close to the other fishing boats; this was based on the observations of real-life data. The full-day fisher trajectory was generated according to the Markov chain of the two states (see Fig. 2) and the movement model, and one independent trajectory was generated every day. The initial location of each fisher in the fishery was randomly generated in the 2-D landscape, and the first state of the day was searching. For simplicity, no among-fisher movement variability was considered.

## Simulation scenarios: with and without circadian behavioural variation

Here, we were interested in the individual differences in the daily timing of switching the circadian state in fish, and we were particularly interested in the repeatability score ($R$) of two behavioural manifestations of fish circadian rhythms (Fig. 1): (i) awakening time and (ii) rest onset time (referred to as minutes from sunrise or sunset, respectively). $R$ assesses the degree of consistency of the behaviours displayed by individuals over time (*Nakagawa & Schielzeth, 2010*) and represents the phenotypic variation that is attributable to individual heterogeneity; additionally $R$ is often used to characterise animal personalities and, in our context, to detect chronotypes (*Alós, Martorell-Barceló & Campos-Candela, 2017*; *Dingemanse & Dochtermann, 2013*; *Stuber et al., 2015*). To test our

hypothesis on how fishing selection acts on this circadian behavioural variation, we simulated two scenarios. In the first scenario (i.e. the real scenario), the fish population showed significant repeatability in the awakening and rest onset times, which generated chronotypes (Fig. 1). Each of the 2,000 fish was randomly assigned an individual mean and standard deviation (s.d.) in the awaking and rest onset times according to the real data published in *Alós, Martorell-Barceló & Campos-Candela (2017)* to generate chronotypes using the function `sample` of the R package (range of individual means of awakening time: 18.2–271 min; range of individual means of rest onset time: −9.3 to 13.4 min, see Fig. 1). In the second scenario, all individuals in the population had an awakening time and a rest onset time with the same normal distribution (mean = 0 min, s.d. = 15 min) to obtain an ecological landscape where chronotypes did not exist (i.e. no real circadian behavioural variation nor between-individual differences were simulated), and all individuals were vulnerable to the fishing fleet. This second simulation scenario was used to confirm that the potential selection gradients obtained in the first scenario were not caused by the indirect selection of other behavioural traits or by other unknown dynamics of the IBM. We initially considered a total number of 133 fishing boats (spatial fishing effort: 11 boats per $km^2$) based on the empirical data found for our target fishery, and the main results are discussed using this relatively realistic scenario (*Alós et al., 2016b*). However, both simulation scenarios were finally run using six fishing pressure scenarios that differed in the number of mobile fishing boats exploiting the ecological landscape. We used a wide range of different fishing effort scenarios to evaluate the strength of the potential selection under different fishing pressures (i.e. two, four, six, eight, 10 and 12 boats per $km^2$). These different fishing pressure values generated increasing exploitation rates that ranged from 24% to 70% of the population, which indicates our conclusions can be interpreted for a wide range of scenarios.

## Model outcomes: exploitation model and estimation of selection gradients

The coupled social-ecological landscapes were simulated, and the encounters between fish and fishers were quantified in the two simulation scenarios under the different fishing pressures described above (Fig. 2 and see movie in SM1). We defined an encounter as successful when (i) the distance between the fish and the fisher was less than 5 m (a reasonable distance to assume visual contact of the bait by the fish) in a given time-step, $n$; (ii) the fish was in a vulnerable state (i.e. active); (iii) the fish had not previously encountered another fisher (emulating harvest with depletion); and (iv) the fisher was in the fishing state. When the four conditions were met, the fish ID was considered as harvested and was removed from the simulation to emulate fishing with depletion. Once the simulated fishing season ended, we characterised the surviving individuals (i.e. the exploited population) in terms of their circadian and spatial behavioural variation. We then estimated the selection gradient ($S$) of the two circadian (awakening time and rest onset) and spatial (radius of the HR and exploration) behaviours. $S$ was computed as the difference between the phenotypic mean trait of the initial population and the mean of the surviving population, and values were mean-standardised ($S_\mu$) to generate a

**Table 1 Properties of the initial and exploited populations and selection gradients.**

| Observed fish chronotypes | Initial (n = 2,000) | | Exploited (n = 650) | | $S_\mu$ | | | |
|---|---|---|---|---|---|---|---|---|
| | Mean | s.d. | Mean | s.d. | Mean | s.d. | CI-low | CI-high |
| Home range size (m) | 203 | 90 | 183 | 79 | **−0.52** | **0.07** | **−0.65** | **−0.37** |
| Exploration (min$^{-1}$) | 0.006 | 0.005 | 0.005 | 0.005 | **−0.22** | **0.03** | **−0.29** | **−0.15** |
| Awakening time (min) | 139 | 73 | 165 | 68 | **0.85** | **0.05** | **0.74** | **0.95** |
| Rest onset (min) | 4 | 7 | 4.2 | 6.8 | −0.002 | 0.019 | −0.039 | 0.035 |
| No fish chronotypes | Initial (n = 2,000) | | Exploited (n = 315) | | | | | |
| Home range size (m) | 204 | 88 | 185 | 83 | **−0.49** | **0.11** | **−0.72** | **−0.28** |
| Exploration (min$^{-1}$) | 0.006 | 0.005 | 0.004 | 0.004 | **−0.36** | **0.05** | **−0.44** | **−0.26** |
| Awakening time (min) | 0.7 | 3.8 | 0.7 | 3.8 | −0.0003 | 0.01 | −0.02 | 0.02 |
| Rest onset (min) | −0.5 | 3.9 | −0.3 | 4 | −0.008 | 0.007 | −0.021 | 0.006 |

Note:
Mean and standard deviation (s.d.) in the initial and exploited populations of the four behavioural traits studied here resulting from the simulation scenario where wild fish chronotypes were simulated and fishing effort was 11 boats per km$^2$ per exploitation day. Mean and s.d. of the mean-standardized selection gradients ($S$) and their confidential interval (CI) resulting from the 1,000 bootstrap iterations. $S$ in bold were considered significant.

normalised measure of selection strength following *Matsumura, Arlinghaus & Dieckmann (2012)*, and to ensure they were comparable with previously reported data on other traits. $S_\mu$ is a measure of selection strength and allows the strength of selection acting on each of the various behavioural traits to be ranked independent of the trait's mean and variance. $S_\mu$ can be interpreted as the elasticity of fitness to trait change. For example, a value of $S_\mu = 0.5$ means that doubling the trait value increases fitness by 50%. We computed the 95% confidence intervals of $S_\mu$ for each behavioural trait by bootstrapping (1,000 iterations) the results of the simulation scenarios developed here using the `boot` function of the R package (*Canty & Ripley, 2017*).

## RESULTS

The first simulation scenario that considered real wild circadian behavioural variation (i.e. observed fish chronotypes) adequately reproduced the existence of chronotypes (Fig. 1). The *R* scores in this scenario were 0.43 [0.37–0.6] for awakening time and 0.45 [0.39–0.6] for rest onset, which were similar to the scores obtained from the real data by *Alós, Martorell-Barceló & Campos-Candela (2017)*. Fish started their activity as late as 400 min after sunrise, and among-individual differences in awakening time were clearly recognisable, enabling the identification of an early-riser chronotype (Fig. 1). In contrast, fish finished their activity within a shorter period (up to 20 min after sunset), but some individuals extended their activity by an average of a few minutes according to the real data (Fig. 1).

The mean and s.d. of the four behavioural traits in the initial and exploited populations are shown in Table 1. In total, 650 individuals survived (exploitation rate = 67.5%) in the simulation scenario, and in general, the exploited population was composed of individuals with later awakening times, similar rest onset times, smaller HRs and slower exploration (Table 1). These results generated significant $S_\mu$ that differed from zero in terms of awakening time (mean $S_\mu = 0.85$), HR size (mean $S_\mu = −0.52$), and exploration

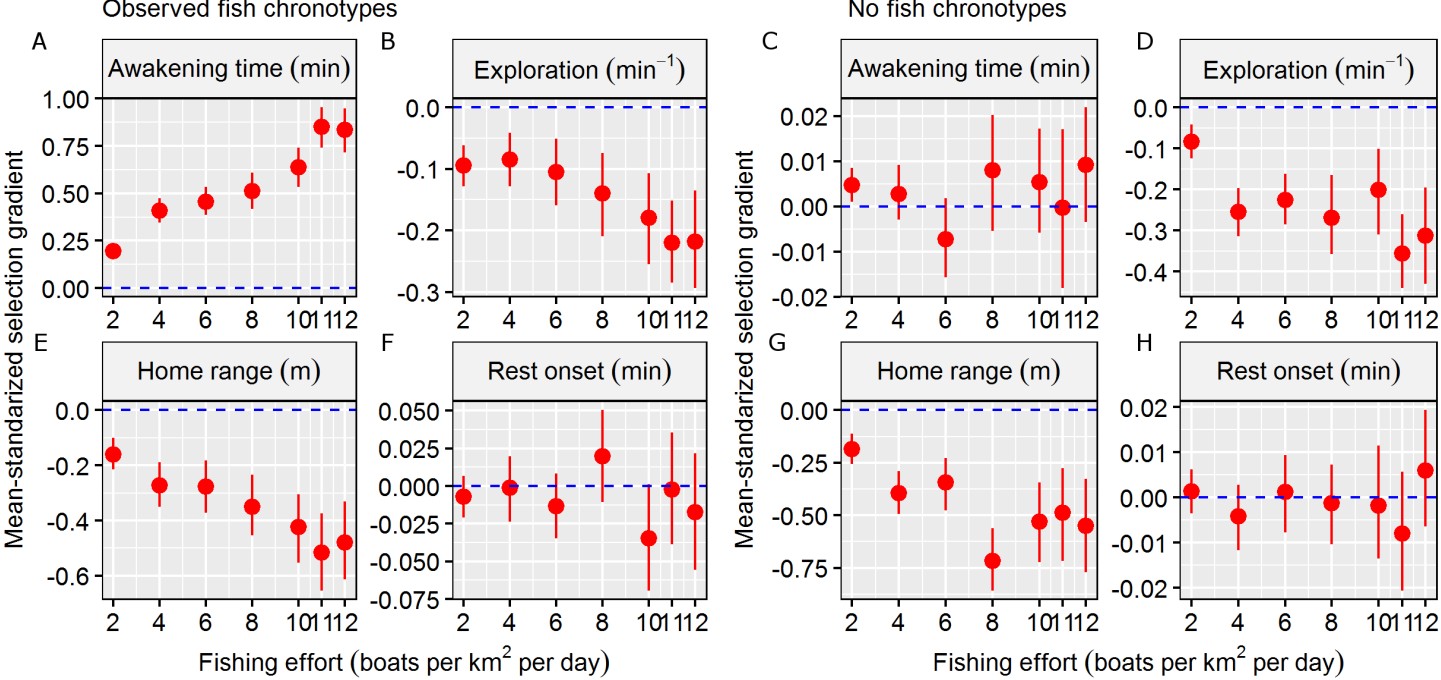

**Figure 3  Fishing selection acting on behavioural traits.** Mean-standardized selection gradients and their confidential interval (CI) obtained in the two simulation scenarios (observed fish chronotypes vs. no fish chronotypes) in a gradient of fishing effort (defined as number of boats per km² per exploitation day) in the four behavioural traits considered: (A and C) awakening time (as min relative to the sunrise), (B and D) the level of the exploration of the home range (as min⁻¹), (E and G) the home range of the individual (defined as the radius of the circular home range in metre) and (F and H) rest onset (as min relative to the sunset). We considered significant mean-standardized selection gradients when the CI didn't overlap with the non-directional selection scenario (plotted as a dashed blue line).

rate (mean $S_\mu = -0.22$) (Table 1). These results were consistent along the simulated gradient of fishing effort, and the strength of significant $S_\mu$ values increased as fishing effort increased (Fig. 3).

In the second simulation scenario, i.e. where no fish chronotypes were simulated, the number of surviving individuals was 315 (exploitation rate = 84.2%). In this case, the exploited population was composed of individuals with similar awakening times and rest onset times and smaller HRs and exploration rates (Table 1). These results generated significant $S_\mu$ different form zero values for only the HR size (mean $S_\mu = -0.49$) and exploration (mean $S_\mu = -0.36$, Table 1), and we discarded significant $S_\mu$ that differed from zero for the circadian behavioural traits (Table 1). These results were also consistent across the simulated gradient of fishing effort (Fig. 3). Therefore, we assumed that the results from the observed fish chronotypes simulation scenario were caused by factors other than circadian behavioural variation.

## DISCUSSION

Circadian behavioural variation have important implications for individual fitness, and many eco-evolutionary trends are dependent on the realised expression of circadian rhythms (*Roenneberg, Wirz-Justice & Merrow, 2003*; *Wicht et al., 2014*); however, very little

information is known about the consequences of fish chronotypes. Here, we found that fishing selection may influence the variation in circadian behaviours by differentially harvesting early-riser chronotypes, and the strength of this selective process is linked to the fishing pressure. We demonstrated these potential consequences of fish chronotypes in exploited environments using a novel social-ecological IBM. IBMs are especially appropriate for formulating and testing emergent population properties from individual processes in predator–prey systems (*Barbier & Watson, 2016*; *Watkins & Rose, 2017*), including fisheries (*Alós, Palmer & Arlinghaus, 2012*). The *R* score or the within-population behavioural variation are classic examples of an emergent population property from individuals, and this value makes IBMs particularly suited to test our hypotheses (*Bell, Hankison & Laskowski, 2009*). In addition, our IBM allowed us to test our working hypotheses using two different ecological simulation scenarios using real data and a wide range of fishing pressure scenarios. Therefore, we feel our approach, although theoretical, properly reproduce some of the potential fitness consequences of circadian behavioural variation in exploited marine environments and provide novel insights in the selective properties of fishing.

The results of the first simulation scenario, i.e. that which used real wild circadian variability, revealed a significant selection gradient in terms of the awakening time. Fish that survived the simulated fishing season were clearly not a random sample of the initial population, and early-riser chronotypes were more prone to capture by the fleet of boats. This finding adds a new variable to the complex concept of the vulnerability of fish to fishing (*Lennox et al., 2017b*). This result was consistent across all fishing pressure levels, suggesting that even in low fishing pressure scenarios (i.e. two boats per km$^2$), fishing selection may influence circadian behavioural traits. In fact, the strength of selection was expected to increase as fishing pressure (i.e. mortality) increased. In contrast, no evidence was found for any selective properties regarding the time of rest onset, which was likely related to the fact that simulated fishing activity mainly occurred during the daytime. In the second scenario, where no wild circadian behavioural variation was simulated, the selection gradient of the awakening time was not significantly different from zero, confirming that chronotypes were major drivers of selection force; this result agrees the results of the first simulation scenario that was based on real-world data.

The potential for eco-evolutionary changes in chronotypes under human pressure has been recently proposed (*Helm et al., 2017*). In fact, *Dominoni et al. (2013)* demonstrated that city European blackbirds, *Turdus merula* began their activity earlier and had faster circadian oscillation than did their forest con-specifics. The results by *Dominoni et al. (2013)* suggested that humans (through artificial lighting) may have selected for individuals by favouring those with large circadian period lengths. In this example, the selective force imposed by artificial lighting acts in the opposite direction than that of our working hypothesis. In our work, the selective force is imposed by the timing of the fishing pressure (Fig. 2); thus, the selective force should favour small foraging periods. What is relevant in this context is that both artificial lighting in the city and fishing pressure in the sea may impose selection gradients in circadian behavioural traits and

may act as eco-evolutionary drivers in wild populations, and this information should be further studied (*Helm et al., 2017*). In addition, our work provides the first evidence that suggests fishing may play a role in the circadian rhythms found in oceans.

Our theoretical selection gradients were mean-standardised, which allowed them to be compared with other traits. First, we found significant selection gradients in the two spatial behavioural traits considered here, indicating selection against large HRs and fast exploration rates. Although both were smaller than the values obtained by the circadian behavioural traits, significant selection gradients were consistent between the two simulation scenarios and across all fishing pressures. Interestingly, the direction of selection was consistent with the empirical selection gradients of hook-and-line recreational fisheries on these spatial behavioural traits, indicating that our IBM was robust (*Alós et al., 2016b*). Moreover, the strength of the obtained selection gradient on awakening time was also stronger when compared with other life history ($S_\mu = 0.66$) and morphological ($S_\mu = 0.29$) traits that have previously been reported (*Hereford, Hansen & Houle, 2004*), although in function of the fishing scenario simulated. In fact, the strength of selection may vary according to the morality pressure, as revealed by the different fishing effort scenarios in our simulations. This fact highlights the relevance of estimating selection gradients in real populations that are exposed to mortality pressure and the importance of using realistic scenarios (i.e. those based on data from the wild). However, our results demonstrated that the potential of selection on circadian behavioural traits certainly exists, and the selection strength could be similar or even stronger than that of previously considered traits.

Although our work is mainly theoretical, we can derive some ecological implications about the selective properties of fishing acting against early-riser chronotypes. Chronotypes are important determinants of reproductive success in birds; for instance, females choose males with early awakening times (*Helm & Visser, 2010*). Assuming this also occurs in fish, one could predict a reduction in the overall reproductive output of a population due to the absence of highly reproductive early-rising males. In addition, fish such as the pearly razorfish play a key role in the food-web by preying on other taxa (*Castriota et al., 2005*) and serving as prey for larger animals, such as dolphins. Thus, a change in the daily timing in a population of pearly razorfish could induce foraging behavioural changes with impact in the lower and upper levels of the food-web. We can also speculate that fishing-induced selection against early risers is currently occurring, and the results observed by *Alós, Martorell-Barceló & Campos-Candela (2017)* are the result of such selective processes. Therefore, we suggest that the ecological consequences of the selective properties acting on circadian behavioural traits are plausible but may already be occurring. In all cases, there is a need to delve into the causes and consequences of fish chronotypes selection, and more empirical work is needed in clarifying the ecological consequences (*Bloch et al., 2013*; *Helm & Visser, 2010*).

The selection gradient is, however, only one component that addresses trait change and derives eco-evolutionary trajectories (*Price, 1970*). The heritability, or the degree of variation in a phenotypic trait in a population caused by genetic variation between individuals, is a key component that can be used to forecast the population-level

consequences of any mortality pressure (including fishing). There is no information on the heritability of chronotypes in marine fish. However, *Helm & Visser (2010)* quantified the heritability of the chronotypes in the great tit, *Parus major* to be 0.86, which is certainly high. In addition, our study is a computational simulation, and it is possible that our results are overestimations because we did not consider other sources of mortality or connectivity; furthermore, we did neither consider other traits that may experience fisheries selection (e.g. size, personality-related behavioural traits, age), nor quantified the fitness in terms of expected reproductive lifetime (i.e. cumulated offspring). The early-life stages of the pearly razorfish are pelagic, and the connectivity of the surrounding non-exploited populations should be integrated to estimate the selection gradients (*Alós et al., 2014*). Therefore, there is a need to provide empirical data to support our predictions and to develop more complex meta-population dynamics that provide a more accurate view of the strength of the selection gradients on the circadian behavioural variation. Next-generation individual-based ecological models that aim to make predictions in a changing world would help in this task by accounting for spatial and temporal resources that merge individual fish and fisher behaviour and bioenergetics with potential micro-evolutionary adaptations (*Ayllón et al., 2016*).

## CONCLUSION

Our work demonstrates that the timing associated with fleet activity may generate significant selection on fish circadian behavioural traits. In fact, the direct selection acting on chronotypes can indirectly be a mechanism of fishing selection on migration or breeding behaviours (*Graham et al., 2017*). Therefore, our work proposes a novel view for understanding the selection properties of fishing acting behavioural traits and generates a list of research needs.

First, we should explore how widespread chronotypes are across fish taxa. Adequate technology and approaches used to measure chronotypes in the wild is certainly available (*Alós, Martorell-Barceló & Campos-Candela, 2017*; *Helm et al., 2017*; *Rattenborg et al., 2017*), and further work should also consider nocturnal species or species that focus their activity during the crepuscular hours to evaluate the generality of our findings.

Second, there is a need to validate our theoretical predictions by performing selection experiments in the wild, where fish are monitored while they are being exploited by real fishers (*Alós et al., 2016b*); additionally, this should include different fleet timing dynamics and fish-fishers behavioural interactions. This future work should also help disentangle the synergistic effects of predation risk and fishing from the potential eco-evolutionary dynamics generated by the existence of circadian behavioural variation.

Third, we should identify the mechanisms behind the expression of wild circadian behavioural variation. Chronotypes are the emergent pattern of the interaction between circadian clocks and the environment, which includes potential light entrainment and responses to predation risk, and their study requires a combination of field and laboratory experiments (*Helm et al., 2017*). In addition, we should explore the plasticity and additive genetic variation (including its heritability) of fish chronotypes to evaluate the

potential for evolution in circadian behavioural traits. In a quantitative genetic way, one potential route would be the exploration of candidate genes and polymorphisms linked to chronotypes, such as the CLOCK or the NPAS2 genes (*Stuber et al., 2016*), and how they are translated across generations (*Helm & Visser, 2010*; *Zhang et al., 2017*).

Fourth, in our previous study, we found chronotypes as an independent axis of activity as fish personality trait (*Alós, Martorell-Barceló & Campos-Candela, 2017*). However, there is a need to extend our research to other fish personality traits, such as boldness, exploration, aggressiveness or sociability (*Conrad et al., 2011*), and their feasible interactions. This would help us understand the role of circadian rhythms in the architecture of behavioural variation in fish. In addition, it would be helpful to explore how chronotypes are correlated with individual growth (productivity) or reproductive success, as it has been done with other behavioural traits (*Biro & Stamps, 2008*).

Once this information and population-based approaches become available, we will be able to forecast the relevance of eco-evolutionary consequences of the wild circadian behavioural variation and how human is affecting it. We hope our work stimulates research and debates on this topic.

## ACKNOWLEDGEMENTS

We thank interesting discussion with Valerio Sabragaglia. We also thank Daniel Ayllon, James Smith and Anssi Vainikka for their constructive and helpful comments and suggestions on our manuscript.

### Funding

This study was funded by the research project Phenofish (grant no. CTM2015-69126-C2-1-R) funded by the Spanish Ministry of Economy and Competiveness. Josep Alós was supported by a JdC post-doc grant funded by the Spanish Ministry of Economy, Industry and Competitiveness (ref. IJCI-2016-27681). Andrea Campos-Candela was supported by a FPU predoctoral fellowship (ref. FPU13/01440) from the Spanish Ministry of Education, Culture and Sports (MECD). The funders had no role in study design, data collection and analysis, decision to publish, or preparation of the manuscript.

### Grant Disclosures

The following grant information was disclosed by the authors:
Spanish Ministry of Economy, Industry and Competitiveness: CTM2015-69126-C2-1-R.
Spanish Ministry of Economy, Industry and Competitiveness: IJCI-2016-27681.
Spanish Ministry of Education, Culture and Sports: FPU 13/1440.

### Competing Interests

The authors declare that they have no competing interests.

## Author Contributions

- Martina Martorell-Barceló performed the experiments, analysed the data, contributed reagents/materials/analysis tools, prepared figures and/or tables, authored or reviewed drafts of the paper, approved the final draft.
- Andrea Campos-Candela analysed the data, contributed reagents/materials/analysis tools, authored or reviewed drafts of the paper, approved the final draft.
- Josep Alós conceived and designed the experiments, performed the experiments, analysed the data, contributed reagents/materials/analysis tools, prepared figures and/or tables, authored or reviewed drafts of the paper, approved the final draft.

## Data Availability

The code (which includes the raw data) is included in the Supplemental Files.

## Supplemental Information

Supplemental information for this article can be found online at http://dx.doi.org/10.7717/peerj.4814#supplemental-information.

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
