# Peer review of "Fitness consequences of fish circadian behavioural variation in exploited marine environments"

_PeerJ, doi:10.7717/peerj.4814_

## Round 0.1 · original submission · Minor Revisions

Please find below three constructive, thorough and overall complimentary reviews for your MS. All reviewers are supportive of publication. They also put forward some useful recommendations that should improve the overall quality of the final manuscript. In particular, there are issues regarding the model assumptions, simulation scenarios and interpretation of findings that need to be addressed prior to publication.

Please note there is also a need for a thorough grammatical revision of the MS (I sympathise as a non-native English speaker myself!).

Congratulations on an excellent study.

·

Basic reporting

This paper is clearly written and easy to follow. The structure conforms to PeerJ standards. The introduction and background are a bit long, so I suggest the authors to shorten the first two paragraphs to get faster to the point: the existence in fish of behavioral syndromes and chronotypes that are linked to fitness processes and thus are subject to selection by fishing. The objectives of the study are clearly stated. The literature is relevant and well referenced. Figures are comprehensive and helpful to understand the properties of the model (Fig. 2) and its performance (Fig. 1), as well as to visualize its predictions (Fig. 3). The included table is also helpful.

The authors provided the code of the model, which was written in R. The script is well structured and annotated (parameters defined and source of their values indicated, commands used to perform the different processes of the model step by step explained). Hence, the code can be reviewed, results checked out and the model could be re-implemented in a different programming platform.

Experimental design

The model and experiment set-up (simulation scenarios) were detailed enough in Methods to be replicated. However, I suggest the authors to organize the Methods in a more standard way (at least within the ecological modeling field): 1) model description, 2) parameterization (this could be included in the previous section), 3) simulation scenarios, 4) model outputs and analysis (e.g., R score, selection gradients, etc). Right now, everything is somehow mixed up, so it is necessary to read the different sections when looking up for specific information. I suggest the authors to use the ODD protocol (Grimm et al. 2006, 2010) to describe the model; it is the most widely used protocol to describe IBMs because it was designed to make descriptions of IBMs easier to read (readers know which section of the ODD they must read to get specific info) and facilitate replication (in case they cannot read the programming language in which the model was written). At any case, those are just recommendations to improve the clarity of the Methods section.

Regarding the model assumptions:

1) If I understood it well, time series of the state of fishers (searching vs. fishing) are randomly defined for the entire simulation, independently of the movement of the fish. Fishers change their state following probability rules that have nothing to do with the movement of the fish. So fishers do not automatically change to fishing state when a fish is close to them (and so it is not harvested). Fishing just happens by chance as there is not a real interaction between agents, they do not show any adaptive behavior. Perhaps these rules are not easy to implement in R? In this case, I suggest the authors to use in the future (not for this study, of course) programming platforms specifically developed to design IBMs, such as NetLogo.

Regarding the simulation scenarios:

2) I am not sure whether the second scenario (all fish with more or less the same circadian rhythms – i.e. no chronotypes) is the best approach to confirm that selection gradients in circadian behavioral traits were caused by chronotypes. From my point of view, this scenario rather confirms that fishing selects on the other two behavioral traits (home range size and exploration rate) irrespective of the existence of chronotypes. For me it is totally clear that fishing is selecting on chronotypes, but if the authors want to confirm that selection gradients in circadian behavioral traits are caused by chronotypes, they should run the model with chronotypes but without implementing any individual variability in the other behavioral traits (to get rid of their potential confounding effect). In this way, they could be sure that fishing selects against early-risers independently of their spatial behavior.

3) The authors have shown that increasing fishing effort increases the strength of selection on behavioral traits, including circadian traits. The authors also indicate that the timing associated to the fleet activity highly influences the strength of selection induced by fishing, but this is not tested. It would be interesting to assess how the strength of selection on circadian traits change with changes in arrival and leaving timing of the fleet to the fishery. Again, just a suggestion.

Validity of the findings

The results are reasonable given the model assumptions and simulation experiments performed (though see comment 2 in previous section). The modeling framework used is certainly the most fitted approach to answer the questions posed by the authors and opens a promising way to push the science on this field forward. The model is parameterized with extensive field data (both in the ecological and social systems modeled) and the model is able to reproduce observed empirical patterns so I feel model predictions are robust.

Additional comments

Definitely a very interesting study. In a previous work, the authors discovered for the first time the existence of chronotypes in fish, and here they show how human activities –recreational fishing- can select against certain chronotypes by using an individual-based approach within an innovative social-ecological system framework. The authors also discuss on the research needs to better understand the selective pressures of human activities on fish behavioral traits.

I provide below some minor comments:

INTRO:

L41-42: positive size-selective fishing not always select for faster life-histories (see Enberg et al.2012)
L49: not ONLY life-history traits
L51: INTEREST instead of "interested"
L53: "On the one hand,..." instead of "In one hand,..."
L56: "On the other hand," instead of "In the other hand,..."
L59: Alós et al. 2016b? No Alós et al. 2016a has been cited previously.
L64: "In consequence," instead of "Because this reason,".
L64-66: please, rephrase the whole sentence.
L65: what do the authors mean with "resident" here? That the fish move only a short distance during the day?
L74: "led" instead of "leaded".
L87: in L82 the authors included the scientific name of the species within brackets, please use only one way of indicating the scientific name, for consistency.
L89-92: this sentence is difficult to understand, please rephrase it.
L93: indicate some examples of fitness processes linked to chronotypes.
L97: selection ON
L97: maybe "leading to impacts on fitness"?
L98-99: what's the impact on fitness caused by earlier activity here?
L98: highlighting THAT urban...
L101: "enhanced encounters"? I assume the authors mean that the number of encounters between fish and fishers are higher just because fish start moving earlier.
L103: based ON relatively... The same in L104 (change" in" to "on").
L109: consequences for? Fitness consequences?

METHODS:

L169: from the base package in R?
L179: add that the values are generated from a normal distribution
L191-193: I guess the authors mean that they combined the two ecological scenarios with six different scenarios of fishing effort.
L203: to a movement MODEL based
L208: type OF animal
L209: delete “recent”, and name the package
L234: something is missing here, perhaps: to the centroid of the AREA WHERE other boats were fishing?
L236: turning angle drawn FROM
L251: in iii), do the authors mean that the fish has not been harvested before by another fisher? Otherwise, if an active fish has had an encounter with a fisher but this wasn’t a fishing state (not harvested then), is then considered not vulnerable to fishing? How long?
L254: SURVIVING individuals
L256: in the two circadian and spatial TRAITS
L264: of the BOOT R-package, provide citation then

RESULTS:

L267-273: I understand these results are to evaluate the model performance, to show how well the model reproduces the patterns observed in the field, with which the model was in fact parameterized. If the model were unable to reproduce such patterns, then model predictions wouldn’t be reliable. Shouldn’t it be described in methods?

DISCUSSION:

L304: in two different ECOLOGICAL SIMULATION SCENARIOS UNDER a wide range of fishing pressure scenarios
L325-326: THROUGH artificial LIGHTING
L327: by artificial LIGHTING
L335: fast EXPLORATION rates
L339-341:
L342: according to the MORTALITY pressure
L353: that OUR results
L355-356: the IBM presented here is not a population dynamics model. It is a specific model designed to test the selective consequences of fishing on chronotypes. So how could these elements help improve the current model?
L356: should be integrated TO ESTIMATE the
L362-364: Has this pattern been observed in fish? I mean, this pattern has been observed in birds, is it applicable to other taxa?
L366: delete “of the populations”
L371: need to DELVE INTO the causes

CONCLUSIONS:

L376-378: difficult to understand what is meant. Please, rephrase.
L376-378: So the simplest management action would be to forbid fishing during the first hours after awakening to avoid the harvest of early-risers?
L386-387: just if a population dynamics model is built…
L393-394: move “are” to the end of the sentence.

·

Basic reporting

I think the overall structure of the manuscript is great, and it generally provides sufficient background and context. However the manuscript contains some poor grammar. The English will need to be improved to ensure this study is comprehensible to an international audience. The abstract is well written, and I expect this has had editing for English where the rest of the manuscript has not. I have edited the first two paragraphs on the attached pdf to highlight these instances, but the remainder requires similar attention.

Experimental design

I liked your approach to testing the role of fishing on chronotype selection. The structure of the IBM looks robust. The question is clear and important, and this study is within the scope of the journal.

One issue I have is that one of your scenarios doesn't seem all that useful. You test two simulations: 1) the chronotype scenario (where individuals show repeatability in behaviour), and 2) the 'no chronotype' scenario (line 172). I don't agree that the second simulation has no chronotypes - all individuals share the same chronotype (they show perfect diurnality) - so it seems like a 'single chronotype' scenario. You state that you use 2) to demonstrate that the selection patterns you saw in 1) are due to the presence of chronotypes, but I think this is a) not really necessary, b) complicated by differences in the exploitation rates of the two scenarios (67% vs 84%). Scenario 2) couldn't possibly show selection for awakening or onset times (Fig 3), so seems more of an internal test to ensure your model shows expected behaviour. To me, a more interesting test would be to vary the strength of the repeatability of the chronotypes in 1) (keeping the same mean inter-individual variation) and plot selection strength versus repeatibility. This would seem relevant as repeatability may be hard to know accurately. If you wish to keep the analyses the way they are, consider labelling 2) a 'single chronotype' scenario, and state that significant selection for scenario 2 in Fig 3) for awakening and onset times was not possible.

Line 105,122 - More info needed on type of fishery (recreational baited hook-and-line)

Line 167 - what were the limits of awakening and onset times you sampled from (using 'sample') - I don't think this info would take up much room, and would save someone having to consult your previous paper.

Line 177 - You call the 'active' and 'resting' chain of states a Markov chain, but I don't see what makes it a Markov chain. It seems the states at each time step is determined for the whole day using a single set of random values - a Markov chain would need random input at each time step? Possibly this stems from a misunderstanding of what is going on here. I assume there are no rest periods during the daytime awakening and onset times? Maybe a little more info here please.

Line 179 - related to the above comment - what is the "one value"? One value for each of the mean and sd?

Line 258 - I would appreciate more info on how S (selection) was calculated. The reference to Matsumura et al. 2012 seems insufficient for some readers to relate variable mortality of chronotypes to the 'S' value in Fig 3. Likewise expand on which "results" (line 264) where bootstrapped.

Line 319 - "the selection gradient in awakening time was not significant" - I think this was unavoidable (see first comment).

In the caption to Figure 2, you mention and map habitat types, and also a visual census of boats. This is the only time they are mentioned and they should be introduced briefly in the methods.

Validity of the findings

Your findings looked generally robust. However, I think you generalise your findings too much ["Therefore, we feel our results are representative of many other fisheries around the globe" (line 307)], without proper acknowledgement (in multiple places) of how species and fishery traits may alter the susceptibility of specific chronotypes to harvest. I think your results are actually due to some quite specific characteristics, and might change considerably given some reasonable real-world conditions: fish can be caught while 'at rest'; different gear types will have different catchabilities for various chronotypes (compare baited hooks with bottom trawls); numerous species show crepuscular and schooling behaviour; different types of habitats can be exposed to different patterns of fisher behaviour (e.g. some reefs are consistently targeted, and small home range individuals may be more vulnerable, instead of large home range individuals as you showed). These are just some characteristics that may alter the strength and direction of selection on chronotypes. This is not to say that what you've done is insufficient, but you need to report clearly that your results are from a specific set of circumstances, and further exploration is needed to identify how this selection acts across a diversity of real species and fishery traits. Some carefully selected words should be added to the abstract and end of the introduction to address this; discussed as caveats in your discussion, and presented as opportunities for further work in your conclusion.

I also think you need to discuss your fishing mortality rate. It seems very high (67.5% in 15 days with 11 boats, or ~12% for 2 boats). I don't believe real-world fishing mortality would ever be this high, and this would surely influence the strength of your selection; you should state that this is much higher than would be expected in reality (unless you find an approximation of fishing mortality F for your model species that supports this). You may have sped up selection greatly in your study with such mortality. If chronotypes are somewhat plastic, then fishing mortality at realistic rates (30% per year?) might show much less impact on chronotypes. You should reword "even in low fishing pressure scenarios" (line 315) with this in mind. You also need to add a caveat to "and its strength could be stronger than other more classical considered traits" (line 345) - this requires validation of realistic fishing mortality rates.

This is optional, but what I think might add addition scope to your findings would be a discussion of why fishing mortality should show different selection pressure on chronotypes than predation mortality. Chronotypes are probably highly selected for by visual (diurnal) natural predators too, so why is fishing different? And might exploiting predators (like sharks) have an equal 'release' of chronotypes in wild systems? Might fishing and predation interact in some ways?

Additional comments

I really like the application of models like this to biological processes, and this seems a perfect application on an IBM. I think it's a great study, and more acknowledgement of species and fishery traits that might influence selection would help promote a productive research area.

Some minor comments:
Line 100 - I think you should change "hypothezied" to "assumed" - there are too many unknows about how species and fishery traits might alter the susceptibility of early riser fish, but using "assumed" shows that those unknowns do not affect the validity of your theoretical study
Line 342 - "morality" should be "mortality"
Line 353 - "ore" should be "our"
Line 386 - what kind of "experiments"? just a few extra words.
Figure 2 - I don't think the blue tracks how up very well - change to white? Consider increasing the size of the red start and end dots.

·

Basic reporting

The manuscripts uses a theoretical but empirically-based approach to predict the evolutionary effects of fishing on fish diel activity. The manuscript idea is novel and important, and in general the manuscript follows a fluent logic, with however, some room for improvement too. The English writing suffers from abundant grammar errors and needs to be proof-read by a native English speaker before the manuscript can be published. The manuscript introduces the relevant literature very well and clearly introduces to the consequences and mechanisms of fishing-induced evolution. All the unclear points are of minor nature and I find the study well conducted and highly interesting, and as such worth publishing with minor edits.

Experimental design

The used IBM is of such technical nature that it is not easy for a random biologist to repeat the work, but given that the model code is provided as well as supplementary material, I find the methods very transparent and according to my competence very well constructed and justified. The main weakness of the IBM is that it is very specific to the model system, and the oceanic currents, for example, appear to represent just a single time period in year 2016. While this adds realism, it also reduces generality. However, I do not see a need to change this as the currents most likely are also in general pretty much like in 2016.

The hypotheses mentioned many times but not specified should be repeated in the methods. Apparently, also a good hypothesis has a yes or no answer, and in your case it might be better to talk about quantitative aims rather than simply answering yes or no whether there is fishing-induced selection on chronotypes. The apparent answer is yes.

Validity of the findings

I find all the presented results valid and having a high impact. The only think I am missing is the answer to your promise to compare your selection gradients to other quantified selection gradients. I expect some comparision to other studies between lines 333 and 346. Please compare your results to other selection gradients caused by fishing (and preferably rod fishing) on any behavioural or life-history traits.

In conclusions I think you should take into account the response of intelligent fishers. If less and less fish are becoming morning-active, more and more fishers will likely start targeting the evening-active fish. Only in special cases should the fishers stick to some tradition and not change their behaviour according to the behavioural change in fish.

Additional comments

I really like the work in general as it touches a novel topic with an interesting model that opens up avenues for many further studies on the topic. However, I think you need to improve the writing (and in few cases the content itself) still a bit.

Here I provide some minor comments that could improve the final version of the manuscript by line numbers:

52: What are these two facts? Please elaborate more the two cited studies.

55: Rather than leading, these differences define the behavioural types

76: Add: temporally consistent

78: It is not very well know if fish sleep. Please introduce the fish sleep theme with a few citations.

101: I would start a new paragraph here with the aims.

114: Please repeat your hypothesis here, it is not entirely here which hypothesis you mean here

118: This is actually a strong assumption and there is studies showing the opposite (see the Monk et al. carp study for example). Make the assumption explicit.

135: Km appears odd why not km?

142: HR = home range I suppose?

165: It would help to repeat the hypothesis here too.

209: Recent is apparently recent, no need to repeat

268: Remind reader that the R scores represent repeatability estimates

282: It would be helpful to call this "second scenario" as control or baseline scenario and introduce its purpose accordingly.

286: Unclear what you mean with "we discarded significant S for the circadian behavioural traits". If the S is significant, you should not discard it. To me it seems the S was practically 0 which indicates what you also conclude: S on chronotypes needs intrinsic variation in them.

292: Instead of starting with a second introduction I would rather start with your main result and achievements in the paper.

298: Nothing is known is wrong as after your results section much is already known already.

302: If this is a classic example, please cite some examples. I am not entirely sure what this sentence means.

309: Please comment how changing daylength in higher latitudes would affect your model interpretation. In many countries, there are also fishers that often fish in evenings or nights, and your morning fishers are globally a special case

342: morality?

398: There is most likely no selection against circadian behavioural traits per se, but selection for or against certain trait values.

---

## Round 0.2 · accepted · Accept

All reviewer comments have been appropriately addressed and the MS is now ready for publication - congratulations on excellent work. Please see a couple of grammatical corrections made in the attached PDF -please incorporate this in your text while in production.

# #